# Impedance Spectroscopic Study of Nickel Sulfide Nanostructures Deposited by Aerosol Assisted Chemical Vapor Deposition Technique

**DOI:** 10.3390/nano11051105

**Published:** 2021-04-24

**Authors:** Sadia Iram, Azhar Mahmood, Muhammad Fahad Ehsan, Asad Mumtaz, Manzar Sohail, Effat Sitara, Syeda Aqsa Batool Bukhari, Sumia Gul, Syeda Arooj Fatima, Muhammad Zarrar Khan, Rubina Shaheen, Sajid Nawaz Malik, Mohammad Azad Malik

**Affiliations:** 1School of Natural Sciences, National University of Sciences and Technology, Islamabad 44000, Pakistan; sadia.iram@sns.nust.edu.pk (S.I.); m.fahad.ehsan@sns.nust.edu.pk (M.F.E.); asad.mumtaz@sns.nust.edu.pk (A.M.); manzar.sohail@sns.nust.edu.pk (M.S.); effat.sitara@sns.nust.edu.pk (E.S.); aqsa.batool@sns.nust.edu.pk (S.A.B.B.); gulniazi_36@yahoo.com (S.G.); 2Department of Materials, University of Manchester, Manchester M13 9PL, UK; malikmohammad187@gmail.com; 3Central Diagnostic Laboratory, Physics Division, PINSTECH, P.O. Nilore, Islamabad 45500, Pakistan; syedaarooj80@yahoo.com (S.A.F.); rubina_shahin_2003@yahoo.com (R.S.); 4Department of Materials Engineering, School of Chemical and Materials Engineering (SCME)-National University of Sciences and Technology (NUST), Islamabad 44000, Pakistan; muhammad.phd@scme.nust.edu.pk (M.Z.K.); sajidnawaz@scme.nust.edu.pk (S.N.M.)

**Keywords:** nickel chalcogenide, AACVD, dielectric behavior, impedance spectroscopy

## Abstract

This research aims to synthesize the *Bis*(di-*iso*butyldithiophosphinato) nickel (II) complex [Ni(iBu_2_PS_2_)] to be employed as a substrate for the deposition of nickel sulfide nanostructures, and to investigate its dielectric and impedance characteristics for applications in the electronic industry. Various analytical tools including elemental analysis, mass spectrometry, IR, and TGA were also used to further confirm the successful synthesis of the precursor. NiS nanostructures were grown on the glass substrates by employing an aerosol assisted chemical vapor deposition (AACVD) technique via successful decomposition of the synthesized complex under variable temperature conditions. XRD, SEM, TEM, and EDX methods were well applied to examine resultant nanostructures. Dielectric studies of NiS were carried out at room temperature within the 100 Hz to 5 MHz frequency range. Maxwell-Wagner model gave a complete explanation of the variation of dielectric properties along with frequency. The reason behind high dielectric constant values at low frequency was further endorsed by Koops phenomenological model. The efficient translational hopping and futile reorientation vibration caused the overdue exceptional drift of ac conductivity (σ_ac_) along with the rise in frequency. Two relaxation processes caused by grains and grain boundaries were identified from the fitting of a complex impedance plot with an equivalent circuit model (R_g_ C_g_) (R_gb_ Q_gb_ C_gb_). Asymmetry and depression in the semicircle having center present lower than the impedance real axis gave solid justification of dielectric behavior that is non-Debye in nature.

## 1. Introduction

Metal chalcogenides (sulfides, selenides, and tellurides) have remarkable technological consumptions in several devices such as solar cell coating, IR detectors, optical conductors, etc. [1,2]. Nickel sulfide (NiS) shows a rather more complex phase diagram as compared to other lanthanide sulfides [3,4]. NiS displays two crystalline arrangements with similarity in their formula but distinct packing of molecules also having unique characteristics [5]. It shows rhombohedral arrangement (millerite) at low temperature whereas, at elevated temperature, NiS depicts ”NiAs-type” symmetry. Within rhombohedral arrangement, five sulfur atoms surround nickel atom in tetragonal pyramidal coordination while its NiAs form shows that every Ni atom is coordinated in octahedral arrangement with a sulfur atom. Semi-metallic behavior with temperature-independent paramagnetism is shown by rhombohedral NiS whereas the hexagonal NiS, which is in the semiconducting antiferromagnetic phase, undertakes transformation to metallic phase [6].

NiS shows 0.5 eV bandgap and p-type semiconducting behavior and has various applications as catalysts [4,7], IR detectors [8], sensors [9], photoelectrochemical [10,11], and solar cells [12] as well in rechargeable lithium batteries behaving as a cathode [13,14]. These features are all dependent on phase, size, morphology, surface properties, and crystal structure which, in contrast, depends entirely on the synthesis approach.

NiS has been synthesized via various techniques including the hydrothermal method [14,15], soft solution-processing [16,17] successive ionic layer adsorption reaction (SILAR) approach [18], as well as laser ablation [19] by different research groups. Different scientists exploited the electrical parameters of NiS synthesized employing different preparation approaches [20,21,22]. P. L. Reddy and coworkers [23] used a green approach to synthesize NiS nanoparticles by utilizing peel of banana extract which acted as reductant. They have fabricated films of nanocomposites made by polyvinyl alcohol (PVA)/NiS using an economical solution casting approach. Furthermore, these films were examined by investigating the dielectric features with varying frequency in the range of 50 Hz–20 MHz along with the temperature ranging from 40–140 °C. Nanocomposites showed a dielectric constant (ε) of 154.55 at 50 Hz and 140 °C. A. Bose and his fellows [24] reported the growth of ~0.6 nm NiS nanosheets along with sodium formica grains, and evaluated the direct and alternating current of NiS nanosheets at 313–473 K, and 10^2^–10^6^ Hz. They concluded that the change in dc resistivity together with the temperature is attributed to parallel arrangement of sodium formica grains and NiS nanosheets. The temperature and frequency dependency on ac conductivity was also carried out by A. Jamil et al. [25] to find out the conduction mechanism of nickel disulfide (NiS_2_) nanoparticles. They utilized the solid-state single-step approach for NiS_2_ nanoparticles. They carried out ac electrical measurements within 300–413 K and observed the depression in both semicircles within the frequency variation between 20 Hz–2 MHz, which indicated the formation of grain boundary phases of NiS_2_ at all examined temperatures. They noticed less hopping polaron conduction at temperatures 300–393 K, while barrier hopping conduction was observed at a temperature greater than 393 K. Electrical properties of Nickel Sulfide thin films prepared by chemical bath deposition were also studied by S. Suresh and coworkers. They have measured ac conductivity which showed that conduction relies on both frequency and temperature [26]. Ultrasonic spray pyrolysis (USP) technique was also used by F. Atay and fellow workers to deposit NiS films onto glass substrates at 300 ± 5 °C. They noticed deviations of temperature-dependent conductivity under the influence of both light and dark conditions and noticed that the conductivity has shown a decrease by illumination [27]. S. Surendran [28] prepared α-NiS particles at different pH conditions (pH 7–pH 14) and noticed the effect of pH on dielectric properties. They observed the change in the optical and electrical properties of the samples by the change in grain size. Likewise, α-NiS obtained at pH 8 showed high dielectric constant (2.3 × 10^4^), and low dielectric loss (13 × 10^3^), in comparison to the sample synthesized at pH 7. O. Balayeva [29] synthesized composites of copper, nickel and functionalized nitrile butadiene rubber (FNBR) polymer by ion exchange method and studied dielectric properties at varied temperature of 26–120 °C in 10^2^–10^6^ Hz frequency range. They did not only measure the electric capacity and resistance of the prepared samples at different frequency, but also noticed the dielectric permittivity, dielectric loss tangent, dielectric modulus, conductivity, relaxation times and Cole-Cole plots. They reported certain destruction process in polymers at 120 °C which affected the interfacial interaction present between the polymer and particles surface.

Previously, we have reported the synthesis of lead sulfide nanostructures from a single molecular precursor *Bis*(di-*iso*butyldithiophosphinato)lead(II) complex by AACVD technique [30]. The current work presents the preparation of *Bis*(di-*iso*butyldithiophosphinato)nickel(II) precursor along with its utilization to deposit NiS nanostructures via aerosol assisted chemical vapor deposition (AACVD) approach, and the impedance spectroscopic analysis of deposited NiS nanostructures. An obvious variance in the shape and appearance of these films of NiS was observed in comparison to those described earlier in the literature from dithiocarbamates [31], di biuret [32], or xanthates [33], or alkyl thiourea complexes [34] of nickel. Different behaviors were shown by each single-source precursor on the deposition of nanostructure by AACVD for the reason that each complex has shown variant decomposition temperature as exhibited by TGA analysis of these precursors.

The major aim of the present study was to grow NiS nanostructure by AACVD of *Bis*(di-*iso*butyldithiophosphinato)nickel(II) complex and also the exploration of its structural, morphological, and dielectric properties. The synthesis of NiS from Bis(di-isobutyldithiophosphinato)nickel(II) complex is novel to the best of our knowledge since this complex has been used for the first time as a single-source precursor to deposit NiS thin films by AACVD.

## 2. Materials and Methods

Chemicals including sodium diisobutyldithiophosphinate, nickel chloride, ethanol, acetone, tetrahydrofuran, and methylbenzene were purchased from Sigma Aldrich, Manchester, UK. These materials endured in purified analytical form and utilized as received.

### 2.1. Preparation of Bis(di-isobutyldithiophosphinato)nickel(II) Precursor

Synthesis of *Bis*(di-*iso*butyldithiophosphinato)nickel(II) complex was accomplished as per the procedure described by Kutchen and co-workers [35]. In a typical procedure, a dropwise addition of a dilute aqueous solution of sodium di-isobutyldithiophosphinate (NaiBu_2_PS_2_) was carried out into the aqueous nickel chloride solution along with continuous stirring. Precipitates thus formed were filtered and dried in a vacuum oven. Finally, crystals of the *Bis*(di-*iso*butyldithiophosphinato)nickel(II) complex were obtained via recrystallization of these precipitates by using methylbenzene/acetone. Calculated elemental analysis (Appendix A) for C_16_H_36_NiP_2_S_4_ (MW = 477.36) indicated C (40.26%), H (7.6%), S (26.86%), P (12.98%) and Ni (12.3%), while experimental values showed C (40.22%), H (7.6%), S (26.97%), P (12.82%) and Ni (12.06%). IR (cm^−1^) (Appendix A): 767 ν (P = S); 722 symm (Ni-S); 1060 asymm (Ni-S); Mass Spectroscopy (Appendix A): M + *m*/*z* = 477 [23] base peak; *m*/*z* = 235 [NiSP[CH_2_CH(CH_3_)_2_]_2_].

### 2.2. AACVD Technique

AACVD assembly charged with *Bis*(di-*iso*butyldithiophosphinato)nickel(II) complex was assembled to accomplish nanostructures deposition procedure. In a 100 mL, two necks round-bottom flask, 0.8 mmol (0.20 g) of *Bis*(di-*iso*butyldithiophosphinato)nickel(II) precursor was added followed by the dropwise addition of Tetrahydrofuran (THF) (10 mL) until it was completely dissolved. The aerosols that were formed inside the solution flask were transported to a reactor tube by joining an inlet of argon gas with a flow regulating Platon gauge. The reactor tube and the flask were connected by reinforced tubing. Six glass slides (approx. 1 × 3 cm) were kept within a tube reactor which was then positioned within a Carbolite chamber furnace. Flask having the complex solution was kept within the heated water bath of a Pifco Ultrasonic Humidifier. Bubbles of complex suspension particles were formed which were then swiped by a carrier gas into a hot-walled reactor’s zone where thin film comprising NiS was deposited after breakdown of *Bis*(di-*iso*butyldithiophosphinato)nickel(II) precursor. The whole procedure was repeated at 3 different temperatures, i.e., 350, 400 and 450 °C under atmospheric pressure with a 200 SCCM argon flow rate for 60 min. In-situ oxidation was prevented by fleeting the argon gas through it for 10 min at all these observed temperatures. NiS films produced were in grey/black color and adhered firmly to glass surface except the film grown at a lower temperature which was very thin, so films grown on 400 and 450 °C were considered for further studies.

### 2.3. Dielectric and Impedance Studies

We have selected NiS obtained at 450 °C (Ni:S, 53.47:46.53) for impedance analysis since it is closer to ideal stoichiometry i.e., least Ni-rich as compared to other deposition obtained at 400 °C (57.89:42.11) i.e., having Ni in excess as revealed by EDX analysis.

After the NiS powder was scratched from the substrate, a hydraulic press was used to prepare NiS pellets with 13 mm diameter 1.3 mm thickness, which is then sintered at 160 °C for 4 h in a furnace. After this, dielectric and impedance analysis of the as-prepared pellets were carried out at room temperature with a varied frequency between 100 Hz–5 MHz over the Precision Impedance Analyzer of Wayne Kerr. 6500 B (5 Commonwealth Ave, Woburn, MA 01801, USA) while using the silver-coated brass electrodes. Fitting of the measured result was performed using ZView software (North Carolina, 3.2 Version). Moreover, various electrical parameters including the dielectric constant (ε′), dielectric loss (ε′), tan loss (tan δ), alternating current conductivity (σ_ac_), complex electric modulus, and the real Z′ were also assessed.

### 2.4. Structure and Morphology

Fourier transform infrared spectroscopy (FTIR) analysis was completed by employing Bruker platinum ATR model Alpha Germany (Bruker, Karlsruhe, Germany). The mass spectrum was noted via micromass Autospec-Q utilizing Micromass Opus software (IET, Mundelein, IL 60060, USA). An electron beam of an impact having 70 eV energy with 10y^7^ Torr was utilized to begin mass fragment formation. Thermogravimetric analysis (TGA) and elemental analysis were performed at the University of Manchester’s microanalytical laboratory while using an organic elemental analyzer of Thermo Scientific Flash 2000 (Thermo fisher scientific, Waltham, MA, USA) along with Seiko SSC/S200 (Seiko, Ginza, Japan) under N_2_. TGA was executed at room temperature to 600 °C at a ramping rate of 10 °C.min^−1^. X-ray diffraction (XRD) pattern was attained using Bruker D8 Advance diffractometer (Bruker, Karlsruhe, Germany) having Cu-Kα source. SEM micrographs were recorded at FEI, xl-30, Scanning Electron Microscope (FEI, Hillsboro, OR, USA). Transmission electron microscopic analysis has been achieved by the Tecnai F30 FEG TEM instrument (FEI, Hillsboro, OR, USA).

## 3. Results

### 3.1. Gravimetric and Spectroscopic Analysis

TGA of *Bis*(di-*iso*butyldithiophosphinato)nickel(II) precursor shows the clean decomposition of the complex in a single step between 220–320 °C at 10 °C.min^−1^ under N_2_ atmosphere (Figure 1). The calculated value (18%) of NiS in the complex is much higher than the residue of NiS (10%) observed in TGA. The lower value of the residue is ascribed to the loss of sulfur from NiS at a higher temperature which is well documented in the literature [32].

XRD pattern of as-prepared nickel sulfide films is presented in Figure 2a. Diffraction peaks indexed corresponded to a predominantly hexagonal phase of NiS synthesized at 400 and 450 °C with JCPDS No. 03-065-3419 and 0-065-0395, respectively. A weak band at 2-theta value of 41 refers to a glass substrate peak. Detailed Rietveld refinement (Appendix A) has been given in Appendix A. Figure 2b represents the hexagonal NiS, having space group, P63/mmc with S atoms being tightly packed at the hexagon’s corner, and Ni atoms filled in the octahedral voids [36].

SEM micrographs of NiS nanostructures at 400 as well as 450 °C are presented in Figure 3a,b. The film settled at 400 °C has displayed spherical ball-shaped morphology with a size ranging from 1 to 2 microns. The films have shown the uneven morphology with some parts showing the formation of clusters comprising of much smaller crystallites. The films deposited at a higher temperature of 450 °C showed the deposition of comparatively uniform morphology of NiS. The morphology consists of much smaller crystallites mostly of uniform size which clustered together to form a continuous network of clusters. This change in morphology is attributed to the difference in deposition temperature which is an important parameter to affect the morphology, and it proves in our case.

EDX analysis (Appendix A)has displayed that films contain both nickel and sulfur in 57.89:42.11 ratio at 400 °C and 53.47:46.53 at 450 °C reactor temperatures respectively. These results are also in good agreement with the study carried out by O’Brien et al. [37], who noticed thin depositions of CdS developed by *bis*(di-*iso*butyldithiophosphinato) cadmium(II) precursor. The EDX pattern gave Cd in slight excess (1.00:0.97). This is because boiling points of heavy metals and sulfur are significantly different resulting in their different vaporization rates.

TEM micrographs of the as-synthesized NiS at both observed temperatures are presented in Figure 4a,b. TEM images represent the spherical ball-shaped morphology of the nanoparticles but with different sizes ranging from 150 to 500 nm. Some of the nanoparticles are isolated but most form the cluster with other nanoparticles. There is no noticeable difference between the images of films deposited at 400 and 450 °C (Figure 4a,b).

### 3.2. Dielectric Constant

Dielectric features of NiS nanostructure were noticed at room temperature from 100 Hz–5 MHz. It can be noticed in Figure 5 that the dielectric constant has exhibited a sharp decline with increasing frequency which subsequently becomes constant at higher frequency region. The higher dielectric constant at a lower frequency can be explained based on Koops phenomenological model and Maxwell-Wagner type polarization. Dielectric materials are supposed to be composed of poor conducting grain boundaries which contribute effectively at low frequencies and good conducting grains [38]. This decrease in dielectric constant value along with increasing frequency is credited to interfacial polarization which states that interfacial dipole does not get sufficient period to line up under applied electric field direction at a higher frequency, therefore no major variation in dielectric constant was detected which exhibited that synthesized NiS has good frequency stability. In general, it occurs because, at a higher frequency, the orientation of dipole is challenging. Dipolar polarization is shown by particles involving a dipole moment which changes their orientation under the impact of the applied electric field [39]. The higher dielectric constant indicates the point that NiS nanostructures behave like a nano dipole in an applied electric field. The higher value of the dielectric constant is observed as compared to NiS thin films reported by Suresh et al. [26].

### 3.3. The Dielectric Loss Behavior

The dielectric loss noticed along with a change of frequency detected on room temperature is depicted in Figure 6. The trend depicted by dielectric loss is almost comparable to the dielectric constant. This observed depression in dielectric loss along with a rise in frequency at room temperature has concluded that dielectric loss relies greatly on the applied field frequency [25]. The higher dielectric loss at lesser frequencies is interrelated to charge lattice defect given by space charge polarization at grain boundaries [26]. This lower frequency region corresponds to high resistive grain boundaries therefore the electrons require high energy for hopping between the charge carriers. Moreover, the presence of the crystal defects and imperfections cause a delayed response of the polarization dipoles for their orientation with the changing ac field direction. Thus, contributed more energy loss at lower frequencies [40].

### 3.4. Tangent Loss Behavior

For dielectric materials, tanδ is the major factor that furnishes knowledge regarding energy loss in an electromagnetic field. Figure 7 represents the tanδ values of the synthesized NiS which demonstrated a higher tanδ value at lesser frequencies and vice versa due to interfacial polarization [25,26]. At a higher frequency region, its value is less than 1 in a wide frequency range from 50 kHz to 5 MHz as shown in the inset of Figure 7. The significance of these low values of tanδ and dielectric constant at high frequencies is in photonic and electro-optic material applications as already reported for molybdenum disulfide MoS_2_ [41].

### 3.5. The Ac Conductivity (σ_ac_)

The ac conductivity can be calculated using the following relation
(1)$$\sigma_{ac\;}=\left[\frac{Z^{\prime }}{Z^{\prime^{2}}+Z^{\prime \prime^{2}}}\right]\times \left(\frac{t}{A}\right)$$
where *t* and *A* are the thickness and the area of the sample respectively. The ac conductivity variation along with altering frequency measured on room temperature is displayed in Figure 8. This type of behavior may be explained by dividing the figure into two separate parts; the first part is the lower frequency portion, and second part is the higher frequency dispersive region. It can be observed from the graph that in the low-frequency region the conductance is almost frequency independent and the conduction mechanism is the same as that of dc conduction. At a particular frequency called the characteristic relaxation frequency, the ac conductivity begins to increase nonlinearly with frequency (dispersive region). This type of conductivity behavior can be explained by the jump relaxation model. According to this model, the lower frequency part has also been related to the successful hopping of ions to the neighboring vacant sites which give rise to long-range translational motion. The second part went through two processes, effective hopping in which after hopping, the ion went to a different position, and unproductive hopping in which hopping ions hopped back to their starting position. During this frequency range competition between these two relaxation processes continued and the increase or decrease in the ratio of effective or unproductive hopping concluded the dispersive area, i.e., the part specifies the two opposing mechanisms which are translational and reorientation hopping processes [42]. This is either the short-range or localized motion of charge carriers. To estimate the dc electrical conductivity and power-law exponent “*s*” the frequency-dependent ac conductivity is fitted by Jonsher’s power law
(2)$$\sigma_{\left(\omega \right)}=\sigma_{dc\;}+A\omega^{s}$$
where $\sigma_{dc}$ represents the dc electrical conductivity and the term $A\omega^{s}$ signifies the frequency-dependent ac conductivity. The value of *s* ≤ 1 corresponds to the short-range translational hopping motion of charge carriers and *s* ≥ 1 suggesting the localized or reorientation hopping mechanism [43]. The fitted parameters are displayed in Table 1.

### 3.6. Electric Modulus

The electric modulus formalism helps to understand the electrical properties of the materials as it can segregate the components having almost similar resistance but different capacitance values. The real and imaginary part of electric modulus can be calculated from impedance data by using
(3)$$M^{\prime }=\omega \;C_{0}\;Z^{\prime \prime }$$
(4)$$M^{\prime \prime }=\omega \;C_{0}\;Z^{\prime }$$
where ω is the angular frequency and $C_{0}={є}_{0\;}A/t$ is the geometrical capacitance, ${є}_{0\;}$ is the permittivity of free space. Change of electric modulus real part, M′, along with altering frequency detected on room temperature is depicted in Figure 9a. Spectra evidenced that M′ is almost zero at a lesser frequency which shows the absence of electrode polarization effect. At a higher frequency above 10^4^ Hz, M′ showed a sigmoid increase along with the increase in frequency that also gained support by lesser mobility ranged charge carriers.

Change of electric modulus imaginary part (M″) along with a change in frequency of as-synthesized NiS is displayed in Figure 9b which gave a well-resolved peak arising at a specific frequency followed by a little asymmetry as shown in the obtained trend. The non-Debye nature of the conduction and relaxation phenomenon and the behavior of the hopping electrons was the basic reason behind this little asymmetry. This slight peak broadening was outcome dispersion of relaxation times by changing time constants. M″ max value gave relaxation frequency f” max which has provided a significant understanding of the dipole relaxation time (*τ*” max). This relaxation peak of M″ is centered at the dispersive region of M′. The low-frequency region below this peak maximum indicates the charge carriers can move over long distances and the high-frequency region above this f_max_ supports the short-range motion of the charge carriers [44]. Figure 9c demonstrates the complex modulus plot of M″(f) vs. M′(f), which supports the fact that the grain boundary has affianced the maximum dimension. The complex electric modulus, defined as the reciprocal complex permittivity, in principle suppresses any low-frequency electrode polarization effects [45,46]. This depressed and asymmetric semicircle is indicative of a non-Debye type of relaxation behavior. Origination of the semicircle from the beginning has proposed, that there is no electrode polarization or some further conductivity phenomenon tangled in this sample. Figure 9c revealed the superposition of two semicircles in high and low-frequency regions suggesting the contribution of both grain and grain boundary relaxation processes respectively. The capacitance of grain and grain boundary can be extracted from the intercept of these semicircular arcs on the real M′ axis.

### 3.7. Impedance Analysis

The measurement of both real and imaginary parts of impedance along with changing frequency noticed on room temperature is illustrated in Figure 10a,b. A higher Z’ value in the lower frequency range was observed, followed by a decrease in Z’ with an increase in frequency and eventually became superposed at the extreme frequency, giving the response that does not depend on frequency. Elevated Z’ expresses more space charge polarization originated through the dielectric system. The value of Z’ was decreased at a high-frequency value. This may be the outcome of release in space charge and also rise in ac conductivity at elevated frequency. Subsequently, Z″ = CR^2^, the imaginary part of impedance along with frequency has augmented the circuit’s utmost resistive component. The existence of a broad and asymmetric peak with the full width at half maxima (FWHM) greater than 1.14 decades suggests the presence of more than one relaxation process [47]. Figure 10c displayed a Nyquist plot which is a deformed and asymmetric semicircular arc with its center positioned below the real Z’ axis. This indicates the departure from the ideal Debye type and the presence of multiple relaxation processes. The impedance data is fitted with an equivalent circuit model comprising of the series combination of (R_g_ C_g_) and (R_gb_ Q_gb_ C_gb_) using ZView software. Constant phase element Q is introduced to address the non-Debye type behavior. The fitted parameters R_g_, R_gb_, C_g_, C_gb_, Q_gb_ and n_gb_ are represented in Table 2.

## 4. Conclusions

*Bis*(di-*iso*butyldithiophosphinato)nickel(II) [Ni(iBu_2_PS_2_)_2_] was efficaciously prepared and confirmed by IR, elemental analysis, TGA, and mass spectrometry. TGA results for [Ni(iBu_2_PS_2_)_2_] revealed that the precursor started breaking down at 220 °C, and accomplished decomposition at about 320 °C. AACVD technique has been employed for depositing films comprising NiS via *Bis*(di-*iso*butyldithiophosphinato)nickel(II) complex at 400 & 450 °C. XRD pattern and SEM/TEM micrographs confirmed the phase and irregular ball-like round morphology of nanoparticles, respectively. NiS nanostructures were distinctive, having a size smaller than 200 nm. Koop’s theory and interfacial polarization effect have helped in understanding the reduction of the dielectric as well as a tan loss with increasing frequency. The ac conductivity (σ_ac_) gave a distinctive leaning with elevating frequency above 10 kHz, this behavior was well explained by Jump Relaxation Model (JRM). Frequency-dependent ac conductivity was fitted by Jonscher’s power law. The value of frequency exponent “s” is less than 1 which suggests the short-range translational motion of charge carriers. Complex modulus plot and impedance analysis have explained the influence of grains as well as grain boundaries headed for capacitance in addition to resistive behavior of polycrystalline medium. To correlate the electrical properties of NiS, an equivalent circuit (R_g_ C_g_) (R_gb_ Q_gb_ C_gb_) has been proposed to fit the impedance plot. The fitted values of R_g_ and R_gb_ show that grain boundaries are more resistive than grains. This systematic study has improved the comprehension of the electrical properties of NiS and widens its application scope in high-frequency devices.

## Figures and Tables

**Figure 1 nanomaterials-11-01105-f001:**
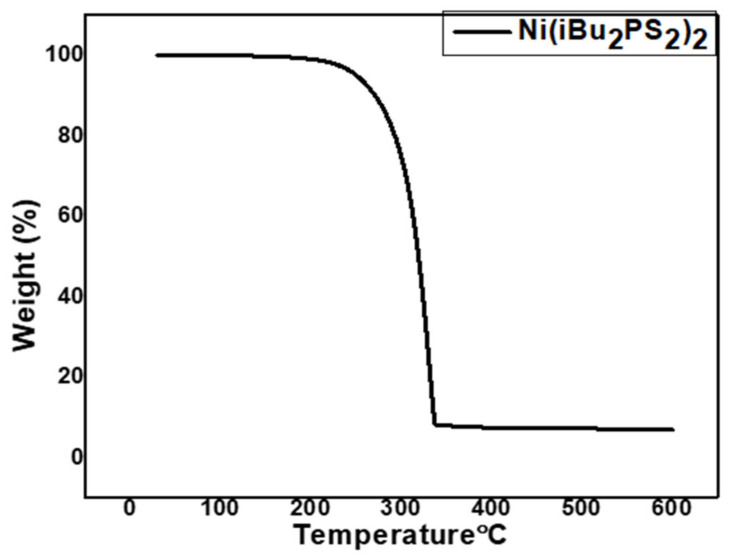
TGA runs of [Ni(iBu_2_PS_2_)_2_].

**Figure 2 nanomaterials-11-01105-f002:**
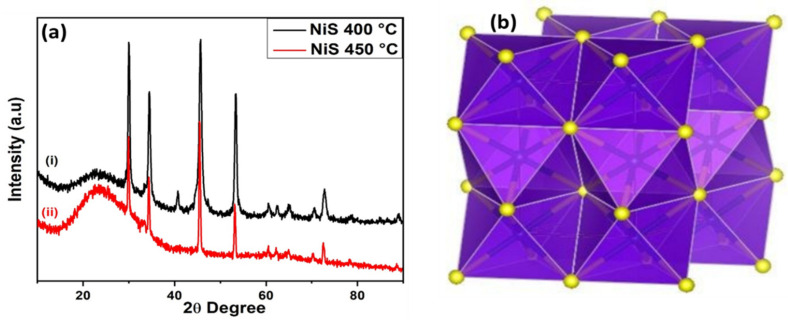
(**a**). XRD spectra of synthesized NiS at: (i) 400 °C; (ii) 450 °C (**b**)**.** Crystal structure of hexagonal NiS with yellow and blue spheres representing the sulfur and nickel atoms [36].

**Figure 3 nanomaterials-11-01105-f003:**
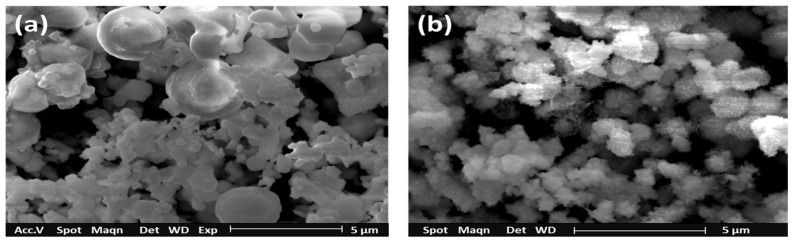
SEM micrographs of NiS nanostructures at: (**a**) 400 °C; (**b**) 450 °C.

**Figure 4 nanomaterials-11-01105-f004:**
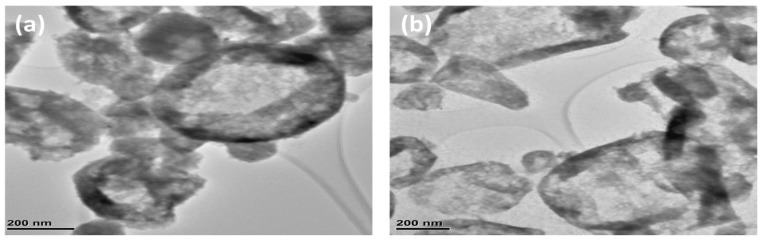
TEM micrographs of synthesized NiS at: (**a**) 400 °C; (**b**) 450 °C.

**Figure 5 nanomaterials-11-01105-f005:**
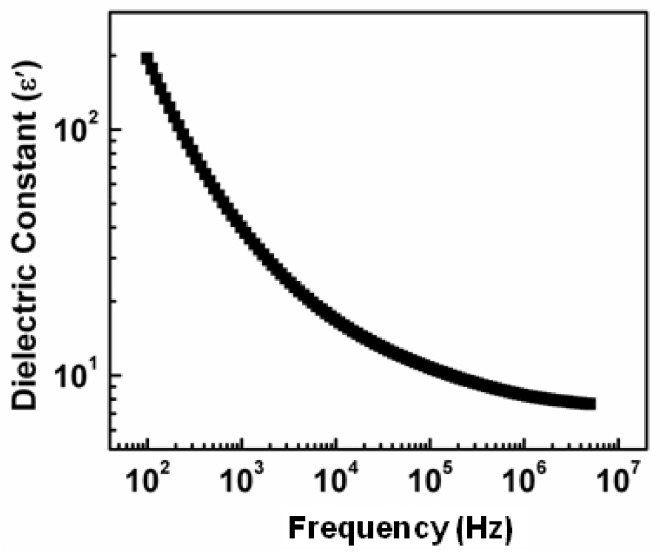
Dielectric constant behavior of synthesized NiS along with a change in frequency.

**Figure 6 nanomaterials-11-01105-f006:**
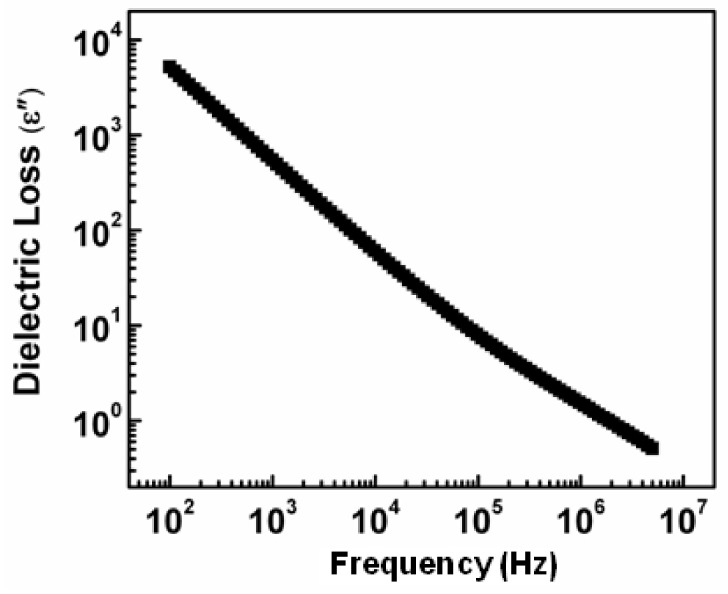
Dielectric loss variation of synthesized NiS along with a change in frequency.

**Figure 7 nanomaterials-11-01105-f007:**
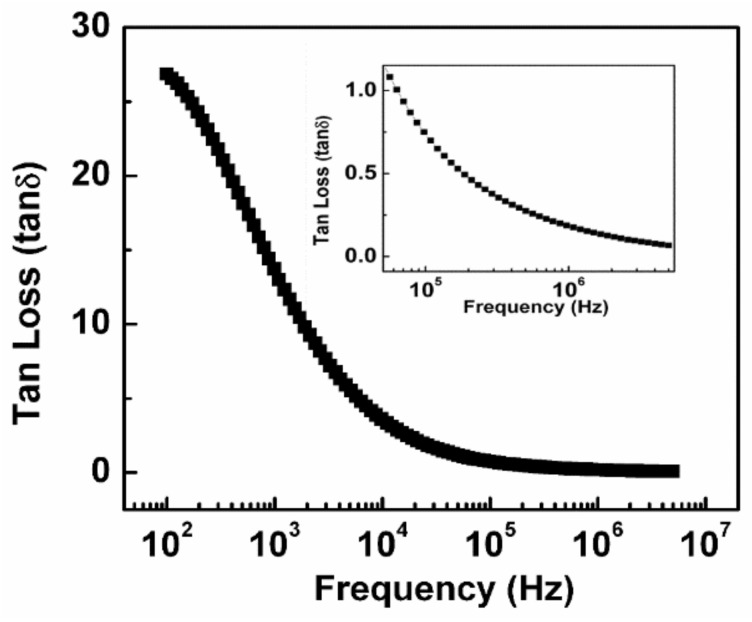
Tangent loss variation of synthesized NiS along with a change in frequency.

**Figure 8 nanomaterials-11-01105-f008:**
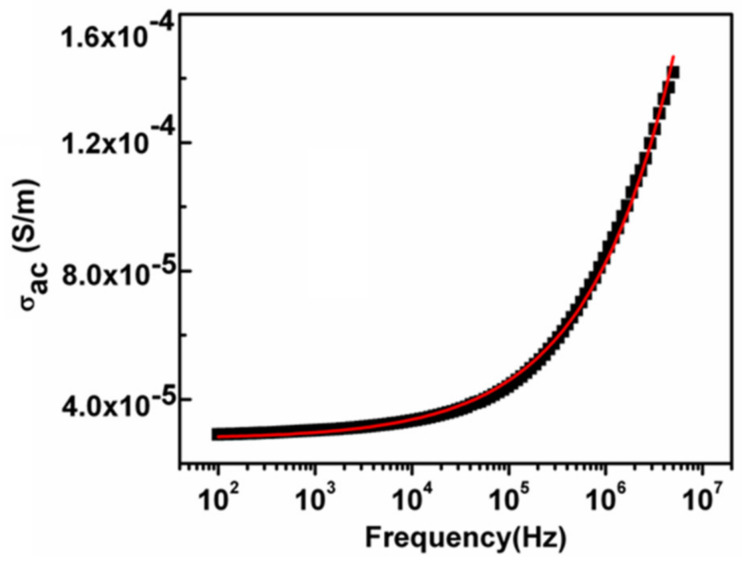
The ac conductivity variation of synthesized NiS along with a change in frequency.

**Figure 9 nanomaterials-11-01105-f009:**
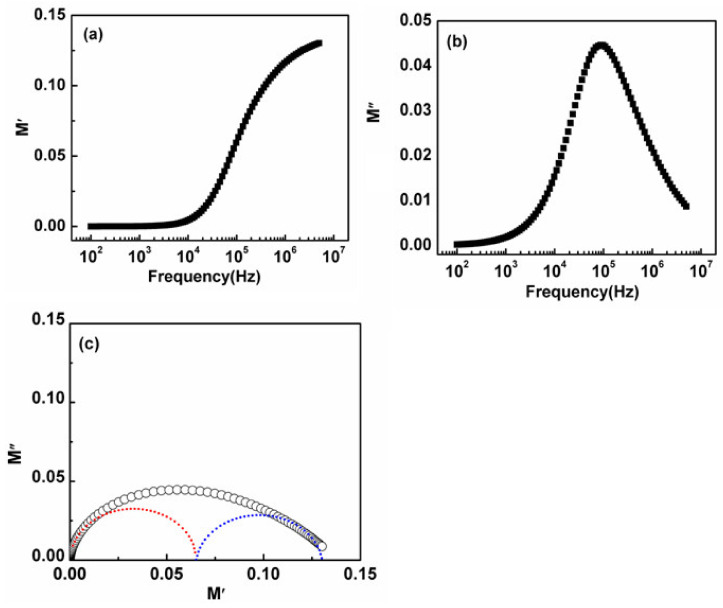
Frequency reliance performance of (**a**) M′; (**b**) M″; (**c**) Complex modulus plot M″ vs. M′ to observe reliability of electrical response going on grain as well as on grain boundaries of NiS.

**Figure 10 nanomaterials-11-01105-f010:**
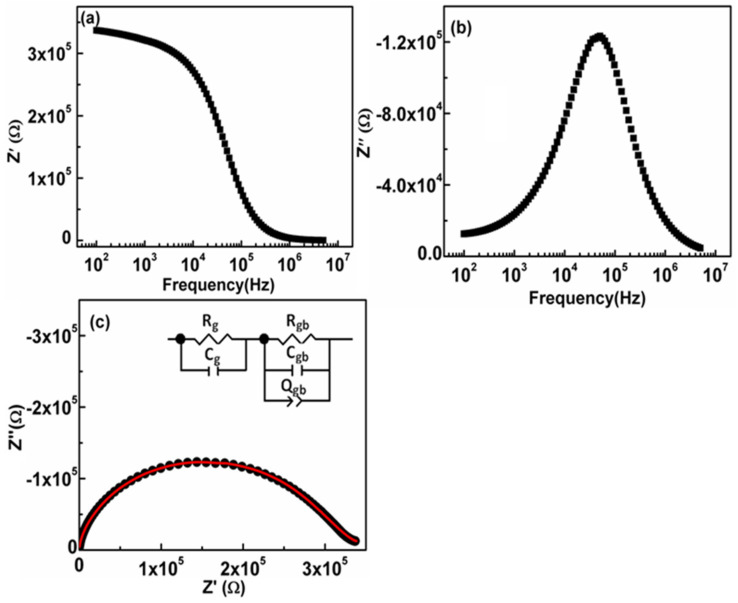
Frequency reliance performance of (**a**) Z′ (**b**) Z″ of impedance formalism; (**c**) Nyquist plot to observe reliability of impedance on grain as well as grain boundaries of prepared NiS.

**Table 1 nanomaterials-11-01105-t001:** The ac conductivity fitting parameters by Jonscher’s power law.

σ_dc_ (S/m)	*A*	*s*
2.76 × 10^−5^	7.2158 × 10^−8^	0.48

**Table 2 nanomaterials-11-01105-t002:** Parameters extracted from ZView fitting of complex impedance plot.

R_g_ (Ω)	C_g_ (F)	R_gb_ (Ω)	C_gb_ (F)	Q_gb_	n_gb_
89679	7.93 × 10^−12^	256380	3.92 × 10^−11^	7.63 × 10^−9^	0.47

## Data Availability

The data presented in this study are available on request from the corresponding author.

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
