# Peer review of "Impedance Spectroscopic Study of Nickel Sulfide Nanostructures Deposited by Aerosol Assisted Chemical Vapor Deposition Technique"

_nanomaterials, 2021, doi:10.3390/nano11051105_

Round 1

Reviewer 1 Report

This version is clearly better than the previous one. But not acceptable, in my opinion.

there are still a lot of corrections to be made.

In particular:

  1. In the answer to Point 3, there is still a lot of typos, such as spaces between values and units. In the new text, presented in red colour, there are typos, which must be corrected.
  2. Line 37, delete “plot by ……..)
  3. Line 75, again error in the frequency, it is not 102 and 106 Hz, of course!
  4. In Table 1, I do not understand what is the last value presented (0.48), as it is not identified in the caption.
  5. Line 277, delete the sentence “These low values of tg …….”, as it is not true, the values are very high. Only for high frequencies, this is true, but at these frequencies, the dielectric constant is very low. So to use this material for energy storage, it is impossible!
  6. Exponents in eq. 1 are not well placed.
  7. The inset of figure 9c) is not clear, actually, it is manipulated! The authors simply delete the data to avoid the error that existed in the last version!
  8. Line 305, actually the authors do not present in table 1 the values of Jonscher model!
  9. Sometimes DC conductivity is sDC, sometimes is s0!
  10. In the Conclusions, the Jump Relaxation Model is cited, but it is not consistent (also in line 292). There is not an evidence of it!

Reviewer 2 Report

Manuscript ID: nanomaterials- 1142548

Deposition of Nickel Sulfide Nanostructured Thin Films: A 4 Potential Material for Impedance Spectroscopic Analysis

by

Sadia Iram, Azhar Mahmood, Muhammad Fahad Ehsan, Asad Mumtaz, Manzar Sohail, Effat Sitara, Syeda Aqsa Batool Bukhari, Sumia Gul, Syeda Arooj Fatima, Muhammad Zarrar Khan, Rubina Shaheen, Sajid Nawaz Malik and Mohammad Azad Malik

The paper concerns a characterization of nickle(II) diisobutyldithiophosphinate complex [Ni(iBu2PS2)], which was used as a precursor for the growing up of the NiS nanostructures on glass substrates by an application of the AACVD technique. [Ni(iBu2PS2)] was characterized by an elemental analysis, the mass spectrometry as well as by IR (I cannot see any results!) and TGA analysis. The polycrystalline film of NIS were analyzed by using the XRD, SEM, TEM and EDX methods. The dielectric response of NiS was investigated at room temperature in the frequency range between 100Hz and 5 MHz.

This is another version of the former paper, a little improved.

  1. The title of the paper is not adequate.
  2. The authors (now there are 13 persons) must know that for different phenomena the different methods are applicable. The sample they analyze exhibits, in my opinion, electric conductivity, which in terms of the frequency dependence of sigmaAC may be described by Jonscher’s equation. In this approach conductivity is frequency-dependent. On the other hand, taking into account the impedance analysis the values of the elements in the equivalent circuit are determined by fitting to the experimental results. In principle, all elements are frequency-independent apart from the constant phase element. This is a completely different approach from Jonscher’s one. In turn, the electric permittivity analysis requires the elements of the equivalent circuit (parallel conductivity and susceptance (capacity)) are frequency-dependent giving, as a result, the Debye-type or Cole-Cole type response. I suggest reading more papers on the electric measurements results. When they have at low frequency the loss tangent over 35 it may mean that they have electronic conductivity (electrons are current carriers). Assuming ion conductivity, (line 294 “According to this model, the lower frequency part has also been related to the successful hopping of ions to the neighboring vacant sites which give rise to long-range translational motion.“ Which ions do they mean?
  3. In their case the analysis of the electric permittivity, e = e’-ie”, is nonsense. They have the DC conductivity value, according to their results, of the order of 1E-5 S/m - this is in the range typical of semiconductors. They do not mention that. Such a large value of DC conductivity makes it impossible to observe any dielectric relaxation processes. They do not discuss the numerical results at all. Nb. Table 1 has no title, the letter “s” (exponent in the Jonscher equation) is omitted in the description.
  4. In line 248 they write “In general, it occurs because, at a higher frequency, the orientation of dipole is challenging.” Which dipole do they mean? The one connected with grain, the polarization of which is connected with its electronic conductivity? Or something else? It would be interesting to have the temperature-dependent results. The conductivity weakly depends on temperature, opposite to the dipolar reorientations. Anyway, they write in line 328 “The non-Debye nature of the conduction and relaxation phenomenon and the behavior of the hopping electrons was the basic reason behind this little asymmetry.” The hopping electrons (not the ions) are consistent with Jonscher’s approach.
  5. By JCPD (line 201) do they mean Joint Committee on Powder Diffraction Standards – JCPDS?
  6. The authors did not describe the electrodes used in dielectric measurements. Did they measure the polycrystalline samples? How were they prepared?
  7. I have some doubts about the sample composition. All over the article, they write NiS but in line221 “EDX analysis has displayed that films contain both nickel and sulfur in 83.88:16.12 221 ratios at 400 °C and 64.47:35.53 at 450 °C reactor temperatures respectively.” Which corresponds to 4:1 and 2:1 composition, respectively. What is the true composition?
  8. Caption of Figure 1. It is better to use “TGA runs” instead of spectra which are commonly used for the energy (wavenumber, wavelength, Hz, eV, etc.) dependence of the results.
  9. Why the conclusions from the SEM and TEM are different?
  10. In conclusion, they write about mass spectrometry. I do not see any results.
  11. The paper should be read by a nature speaker. Some sentences seem unclear.

Reviewer 3 Report

Authors have addressed all issues raised by the referee. It can be accepted now.

Reviewer 4 Report

1

The authors have made significant effort to improve their manuscript. The revised text has reached the standards  for publication. The authors should make an addition that strengthens their work:  Lines 339-340, “The complex electric modulus, defined as the reciprocal complex permittivity, in principle suppresses any low-frequency electrode polarization effects [Mat. Chem. Phys.  232, 319-324 (2019) ; Mat. Chem, Phyhys. 2, 140 (2019); J. Phys. D: Appl. Phys. 48, 285305   (2016) ]”.  I believe this statement and citations justify the selection of the electric modulus as a powerfull  experimental tool.

Round 2

Reviewer 1 Report

Please see the file uploaded!

Reviewer 2 Report

Manuscript ID: nanomaterials- 1142548

Deposition of Nickel Sulfide Nanostructured Thin Films: A 4 Potential Material for Impedance Spectroscopic Analysis

by

Sadia Iram, Azhar Mahmood, Muhammad Fahad Ehsan, Asad Mumtaz, Manzar Sohail, Effat Sitara, Syeda Aqsa Batool Bukhari, Sumia Gul, Syeda Arooj Fatima, Muhammad Zarrar Khan, Rubina Shaheen, Sajid Nawaz Malik and Mohammad Azad Malik

The paper concerns a characterization of nickle(II) diisobutyldithiophosphinate complex [Ni(iBu2PS2)], which was used as a precursor for the growing up of the NiS nanostructures on glass substrates by an application of the AACVD technique. [Ni(iBu2PS2)] was characterized by an elemental analysis, the mass spectrometry as well as by IR (I cannot see any results!) and TGA analysis. The polycrystalline film of NIS were analyzed by using the XRD, SEM, TEM and EDX methods. The dielectric response of NiS was investigated at room temperature in the frequency range between 100Hz and 5 MHz.

This is another version of the former paper, a little improved.

  1. The title of the paper is not adequate.
  2. The authors (now there are 13 persons) must know that for different phenomena the different methods are applicable. The sample they analyze exhibits, in my opinion, electric conductivity, which in terms of the frequency dependence of sigmaAC may be described by the Jonscher’s equation. In this approach conductivity is frequency-dependent. On the other hand, taking into account the impedance analysis the values of the elements in the equivalent circuit are determined by fitting to the experimental results. In principle, all elements are frequency-independent apart from the constant phase element. This is a completely different approach from Jonscher’s one. In turn, the electric permittivity analysis requires the elements of the equivalent circuit (parallel conductivity and susceptance (capacity)) are frequency-dependent giving, as a result, the Debye-type or Cole-Cole type response. I suggest reading more papers on the electric measurements results. When they have at low frequency the loss tangent over 35 it may mean that they have electronic conductivity (electrons are current carriers). Assuming ion conductivity, (line 294 “According to this model, the lower frequency part has also been related to the successful hopping of ions to the neighboring vacant sites which give rise to long-range translational motion.“ Which ions do they mean?
  3. In their case the analysis of the electric permittivity, e = e’-ie”, is nonsense. They have the DC conductivity value, according to their results, of the order of 1E-5 S/m - this is in the range typical of semiconductors. They do not mention that. Such a large value of DC conductivity makes it impossible to observe any dielectric relaxation processes. They do not discuss the numerical results at all. Nb. Table 1 has no title, the letter “s” (exponent in the Jonscher equation) is omitted in the description.
  4. In line 248 they write “In general, it occurs because, at a higher frequency, the orientation of dipole is challenging.” Which dipole do they mean? The one connected with grain, the polarization of which is connected with its electronic conductivity? Or something else? It would be interesting to have the temperature-dependent results. The conductivity weakly depends on temperature, opposite to the dipolar reorientations. Anyway, they write in line 328 “The non-Debye nature of the conduction and relaxation phenomenon and the behavior of the hopping electrons was the basic reason behind this little asymmetry.” The hopping electrons (not the ions) are consistent with Jonscher’s approach.
  5. By JCPD (line 201) do they mean Joint Committee on Powder Diffraction Standards – JCPDS?
  6. Authors did not describe the electrodes used in dielectric measurements. They measured the polycrystalline samples? How were they prepared?
  7. I have some doubts about the sample composition. All over the article, they write NiS but in line221 “EDX analysis has displayed that films contain both nickel and sulfur in 83.88:16.12 221 ratio at 400 °C and 64.47:35.53 at 450 °C reactor temperatures respectively.” Which corresponds to 4:1 and 2:1 composition, respectively. What is the true composition?
  8. Caption of Figure 1. It is better to use “TGA runs” instead of spectra which is commonly used for the energy (wavenumber, wavelength, Hz, eV, etc.) dependence of the results.
  9. Why the conclusions from the SEM and TEM are different?
  10. In conclusion they write about mass spectrometry. I do not see any results.
  11. The paper should be read by a nature speaker. Some sentences seem unclear.

Author Response

Dear Reviewer

The report that you have attached is similar to that we have already answered, if you have other questions, please send us so that we can respond within time. Thanks

Round 3

Reviewer 1 Report

Please see the file.

Reviewer 2 Report

Manuscript ID: nanomaterials-1142548

 Impedance Spectroscopic Study of Nickel Sulfide Nanostructures Deposited by Aerosol Assisted Chemical Vapor Deposition Technique  

by

 Sadia Iram, Azhar Mahmood, Muhammad Fahad Ehsan1, Asad Mumtaz, Manzar Sohail, Effat Sitara, Syeda  Aqsa Batool Bukhari, Sumia Gul, Syeda Arooj Fatima, Muhammad Zarrar Khan, Rubina Shaheen, Sajid Nawaz Malik and Mohammad Azad Malik

Regretfully I have still some remarks.

The Debye-like dielectric response regards the complex value of the dielectric permittivity, which is determined from the admittance value (Y* = G + jwC = 1/Z*, w = omega – radial frequency), not to the impedance value. When we take into account the parallel circuit (G, Cp = const) it gives a semicircle in the (-Z”, Z’) plane. In the Debye-like model G and Cp are frequency-dependent! The statement in the Abstract should be changed.

Fig. 9 c(M” vs. M’) is nonsense. If we assume that the grain and grain boundaries relaxation processes are independent the corresponding semicircles cannot cross.
